

# Flux-tunable hybridization in a double quantum dot interferometer

Christian G. Prosko[1], Ivan Kulesh[1], Michael Chan[1], Lin Han[1], Di Xiao[2],
Candice Thomas[2], Michael J. Manfra[2,3,4], Srijit Goswami[1*] and Filip K. Malinowski[1]

**1** QuTech and Kavli Institute of Nanoscience,
Delft University of Technology, Delft, The Netherlands
**2** Department of Physics and Astronomy, Purdue University, West Lafayette, USA
**3** School of Materials Engineering, Purdue University, West Lafayette, USA
**4** Elmore School of Electrical and Computer Engineering,
Purdue University, West Lafayette, USA

⋆ S.Goswami@tudelft.nl

## Abstract

Quantum interference of electron tunneling occurs in any system where multiple tunneling paths connect states. This unavoidably arises in two-dimensional semiconducting qubit arrays, and must be controlled as a prerequisite for the manipulation and readout of hybrid topological and parity qubits. Studying a loop formed by two quantum dots, we demonstrate a magnetic-flux-tunable hybridization between two electronic levels, an irreducibly simple system where quantum interference is expected to occur. Using radio-frequency reflectometry of the dots' gate electrodes we extract an interdot coupling exhibiting oscillations with a periodicity of one flux quantum. In different tunneling regimes we benchmark the oscillations' contrast, and find their amplitude varies with the charge state of the quantum dots. These results establish the feasibility and limitations of parity readout of qubits with tunnel couplings tuned by flux.

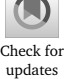

# 1   Introduction

Magnetic fields impart a phase on electron wave functions, leading to constructive or destructive interference between different electron trajectories. This manifests in commonly observed phenomena such as the Aharonov-Bohm (AB) effect and weak localization [1]. Similarly, confined quantum systems where only a few states are coupled to each other can exhibit interference [2–6], for example due to interference of phases imparted by magnetic fields on the couplings [7]. To date however, the phase of tunnel couplings between discrete fermionic levels has never been directly measured. This is particularly relevant for several kinds of semiconductor and hybrid semiconducting-superconducting qubits formed with quantum dots (QDs). QDs are a fundamental component of topological qubits based on Majorana bound states [8–12] as well as spin qubits [13]. They are also naturally suited for quantum simulation [14]. Since measurement-based topological qubits are typically composed of multiple QDs connected in a loop, their hybridization is sensitive to the magnetic flux through the loop because it modulates the tunnel couplings' phases, causing interference [7]. Crucially, this flux-dependent tunneling is a prerequisite for the readout and manipulation of these qubits and for tests of Majorana fusion rules [10–12, 15, 16]. In both situations, the tunneling strength must be adjusted with magnetic flux to maximize measurement sensitivity. Meanwhile, tunneling may depend on flux in two-dimensional QD arrays for quantum processors [17, 18] or quantum simulation [5, 19, 20], since coherent tunneling can occur across chains of QDs [21]. This highlights the importance of understanding and accounting for this effect. Additionally, it has been proposed that new types of semiconducting qubits could exploit flux-tunable couplings to implement gate operations and noise-protected readout schemes [22–24]. Currently, coupling between dots is typically controlled solely electrostatically with gate voltages [25, 26], and an understanding of how magnetic flux affects tunneling amplitudes is lacking.

Motivated by this, we probe quantum interference in the irreducibly simple case of tunneling between two electronic levels in a loop formed by two QDs. Radio-frequency (RF) gate reflectometry is sensitive to tunnel couplings between QDs [27–36], and is a prominent candidate for scalable readout of semiconductor and topological qubits [10–12, 15, 37]. We therefore employ it to quantify the interdot coupling as a function of magnetic flux, and demonstrate a flux-tuned hybridization between electron levels. The specific charge and therefore quantum state of the QD system strongly affects the tunnel coupling and the oscillation amplitude. Importantly for gate reflectometry, the relation between tunnel couplings and measured signal is

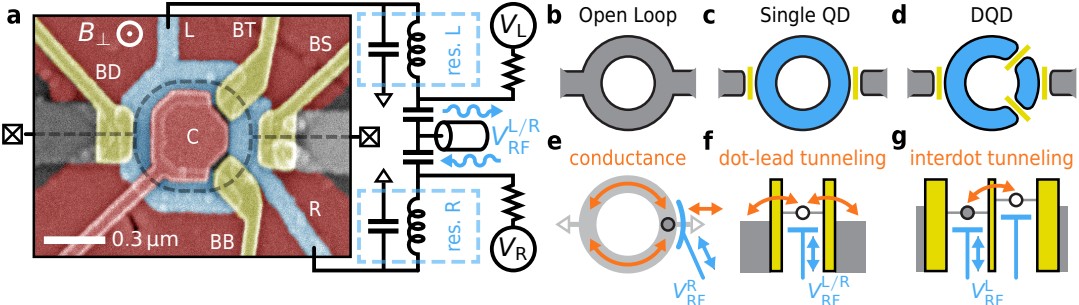

Figure 1: Experimental design and device configurations. **(a)** False-color electron micrograph of a nominally equivalent device to the one measured, and a schematic of the resonator circuit. The device may be tuned by depletion (red) and barrier (yellow, labeled) gate electrodes into an open AB loop, a ring-shaped QD, or a DQD with QDL and QDR chemical potentials tuned by plunger gate voltages $V_L$ and $V_R$ (blue, labeled), schematized in **(b)**, **(c)**, and **(d)**, respectively. Outer and inner depletion gates have $-2\,\text{V}$ and $V_C = -3\,\text{V}$ applied respectively to form a conducting loop unless otherwise specified, illustrated by a dashed line. **(e-g)** Coupling of the resonator voltages to electron tunneling and transport for the three configurations depicted in **(b-d)**. The investigated transport mechanisms which couple to the oscillating resonator voltage (blue) are described with orange text and arrows. For the single **(f)** and double QD **(g)** configurations, we use a chemical potential illustration to show the oscillating resonator voltage coupling to tunneling events (orange arrows). For the open loop **(e)**, its RF conductance dominates the resonator signal. For single and double QDs, incoherent tunneling with the leads has capacitive contributions from tunneling capacitance and dissipative contributions from charge relaxation. In addition, interdot tunneling in a DQD quantifiably translates into a quantum capacitance loading the resonator.

nonlinear [34]. Therefore, contrary to expectation [15], we find that readout fidelity of qubits with their state information encoded in a flux-tuned tunnel coupling may be optimal for weak coupling between the involved QDs.

This manuscript is organized as follows: In Sec. 2, we describe the device fabrication procedure as well as its configurability into an open loop, a quantum ring, or a double quantum dot (DQD). Phase-coherence of electron transport through the device is then established in Sec. 3 in two ways. First, we measure the AB effect manifesting in both DC conductance and RF reflectometry of the open loop. Second, we tune the device into a large loop-shaped QD, and measure $h/e$-periodic oscillations of its addition energy with flux [38,39], where $h/e$ is the single-electron flux quantum. This QD exhibits a consistently finite excitation energy despite having an approximate circumference of $1.4\,\mu\text{m}$ inferred from the oscillations' periodicity. The main result of the manuscript is then presented in Sec. 4, where we demonstrate a flux-tunable tunnel coupling between the levels of two quantum dots arranged in a loop and assess limitations of this tunability in Sec. 5. Lastly, in Sec. 6 we consider implications of these results for future applications to semiconducting and hybrid superconducting qubits.

**Note added** During the review process of this article, a manuscript applying related measurement techniques in a hybrid superconducting-semiconducting device was posted online [75].

## 2 Device overview

To fabricate a device capable of forming a ring-shaped DQD, we use a ternary $InSb_{0.86}As_{0.14}$ two-dimensional ternary electron gas (2DEG) grown as in Ref. [40]. The device (Fig. 1(a)) consists of three Ti / Pd gate layers patterned on the 2DEG, each separated by 20 nm of deposited $Al_2O_3$ dielectric. Charge is confined to an annular ring geometry by applying voltages to deplete carriers below the outer and inner depletion gates (red). The voltage on the inner depletion gate $V_C$ also serves to tune the chemical potential of the entire ring. Voltages $V_{BS}$, $V_{BD}$, $V_{BT}$, and $V_{BB}$ on the barrier gates (yellow) define a large curved QD and a smaller QD (denoted QDL and QDR, respectively), while voltages $V_L$ and $V_R$ on the plunger gates (blue) control their chemical potentials. Specifically, $V_{BS}$ and $V_{BD}$ form tunnel barriers between the QDs and lead reservoirs, while $V_{BT}$ and $V_{BB}$ tune the individual interdot couplings between the QDs via each arm of the loop. Two additional unlabeled accumulation gates (gray) control charge density in the exposed 2DEG between the QDs and Al contacts. Note that when gates in higher layers overlap with gates in lower layers, their applied voltage in this region is screened by the lower metallic gate. Hence, only the region of the gate separated from the 2DEG by dielectric significantly tunes the 2DEG chemical potential. Additional details of the fabrication may be found in Appendix A.

By appropriately tuning gate voltages, the device can be continuously tuned between an open loop, a loop-shaped QD, and a DQD (Figs. 1(b-d)). Measurements on the former two configurations enable us to verify that electron transport is phase-coherent over the ring circumference, and that the ring as a whole supports a single extended electron state. The DQD configuration represents a minimal system in which interference of tunneling between two electron states can occur, as we will demonstrate.

Both plunger gates controlling QDL and QDR are bonded to resonators formed by NbTiN spiral inductors with 420 nH and 730 nH inductance and their parasitic capacitances, leading to resonance frequencies of approximately 400 MHz and 315 MHz, respectively [41]. We measure $V_{RF}^L$ and $V_{RF}^R$: the signal reflected from the resonator connected to gate L or R upon applying a voltage excitation near their resonance frequencies. This complex amplitude depends on the capacitance associated with resonant tunneling and losses from dissipative transport. The former results in a frequency shift of the resonator $\Delta f_0^L$ or $\Delta f_0^R$, while the latter reduces its quality factor [27, 33, 37]. The low-power signals reflected by the device are amplified by a high-electron-mobility transistor at 4 K and measured with a vector network analyzer or ultra-high-frequency lock-in amplifier to produce $V_{RF}^L$ and $V_{RF}^R$, see Fig. 1(a). Using frequency multiplexing [41], both quantities can be measured simultaneously. Measurements are performed at the approximately 20 mK base temperature of a dilution refrigerator.

In each of the three measurement configurations displayed in Fig. 1(b-d), properties of the device are readily measured using RF reflectometry of resonators connected to gates L or R. The reflectometry signal is sensitive to the RF admittance of the device [37]. In the case of an open loop, the resonator on gate R probes the RF conductance of the loop in series with its gate capacitance, depicted in Fig. 1(e). The device admittance is dominated by high frequency conductance of electrons traveling around the loop and into the leads in this case (orange arrows), such that the resonator signal arises primarily due to changes in the resonator's internal quality factor. When tuned into a single loop-shaped QD, both gates L and R tune its chemical potential. Hence, their coupled resonators are sensitive to tunneling effects between the QD ring and the leads. A chemical potential diagram of this coupling is shown in Fig. 1(f). Relaxation events in the form of electrons tunneling between the QD and the leads out of phase with the oscillating gate voltage load the resonator reactively with tunneling capacitance and dissipatively with Sisyphus resistance [37, 42–44]. Through these signal contributions, Coulomb resonances of the QD are measurable since they lower both the

resonator frequency and its quality factor. Finally, when tuned into a loop-shaped DQD, the gate resonators' signals are sensitive to interdot tunneling, depicted in Fig. 1(g). In particular, a substantial interdot tunnel coupling manifests in a purely reactive admittance arising from quantum capacitance [31,33,37], which can be used to directly measure the tunnel coupling [34]. Hence, the measurement signal arises almost entirely from a frequency shift of the resonator due to the additional quantum capacitance.

## 3 Phase-coherent loop and quantum ring

We begin by verifying the electron phase coherence in our device manifested by the AB effect. To form an open loop without QDs, we set all accumulation, plunger, and barrier gates to positive voltages to remove potential barriers. Fig. 2(a) presents the four-terminal conductance $G$ and response of the right gate R resonator as a function of the out-of-plane field $B_\perp$. Oscillations of conductance in flux with a periodicity of $h/ne$ for integer $n$ are expected, depending on how many times an electron can travel around the loop while maintaining a coherent phase [1]. The resonator is sensitive to dissipative transport in the loop despite being capacitively coupled, manifesting as a reduction of the resonator's quality factor. Matching AB oscillations and higher harmonics are prominent in both $G$ and the depth of the minimum in the reflection coefficient of the gate R resonator on resonance [45]. We observe a varying $\phi_0 \equiv h/e$ and $h/2e$ flux periodicity consistent with the expected bounds on area based on the lithographically defined 180 nm and 320 nm inner and outer radii of the loop. This suggests a phase coherence length at least on the order of a micron, based on the inferred circumference of the loop.

To investigate if the entire ring can support an extended electronic state, we continue by tuning the open loop into a large ring-shaped QD. The electron eigenstates of a sufficiently thin ring are angular momentum states with energies quadratic in flux, centered at integer multiples of $h/e$. By virtue of the Pauli exclusion principle, the highest unoccupied electron state is expected to exhibit a zig-zag like pattern in energy with an $h/e$ flux periodicity, illustrated in Fig. 2(b). When the quantum ring forms a QD coupled to leads, this results in analogous kinked oscillations of the dot's addition energy—its spacing between Coulomb resonances—as a function of chemical potential [38,46].

To form such a quantum ring, we lower $V_{BS}$ and $V_{BD}$ to form tunnel barriers (Figs. 1(c) and (f)), and tune the QD's chemical potential with $V_C$. Both gate L and gate R's resonators are sensitive to tunneling between the dot and surrounding leads, since $V_L$ and $V_R$ tune the ring's chemical potential. To project each complex resonator signal into a single real quantity, we calculate the absolute distance of it from the Coulomb blockade signal, denoted $\tilde{V}_{RF}^L$ or $\tilde{V}_{RF}^R$ (See Appendix B). Since both resonators are measured simultaneously in this case, we normalize the resulting magnitudes and sum them for measurements of this QD. In this regime, the large QD exhibits a finite level spacing as demonstrated by the gapped excitation lines visible in Coulomb diamond measurements shown in Fig. 2(c). Moreover, we observe $h/e$-periodic oscillations of the addition energy as the magnetic flux is swept with zero applied bias in Fig. 2(d), consistent with expectations for a quantum ring [38,39]. Though the oscillations are highly irregular, the peak positions and signal strengths' average Fourier transform shows a clear peak at an $h/e$ period of 27 mT, shown in the inset. This corresponds to a circumference of 1.4 µm assuming the ring is circular. Deviations from a regular zig-zag pattern in the addition energy may arise when the ring is not perfectly one-dimensional, such that radial degrees of freedom contribute to its wave function. Potential irregularities along the ring's perimeter and effects of spin-orbit coupling also can cause the more complex oscillations in its addition energy [47].

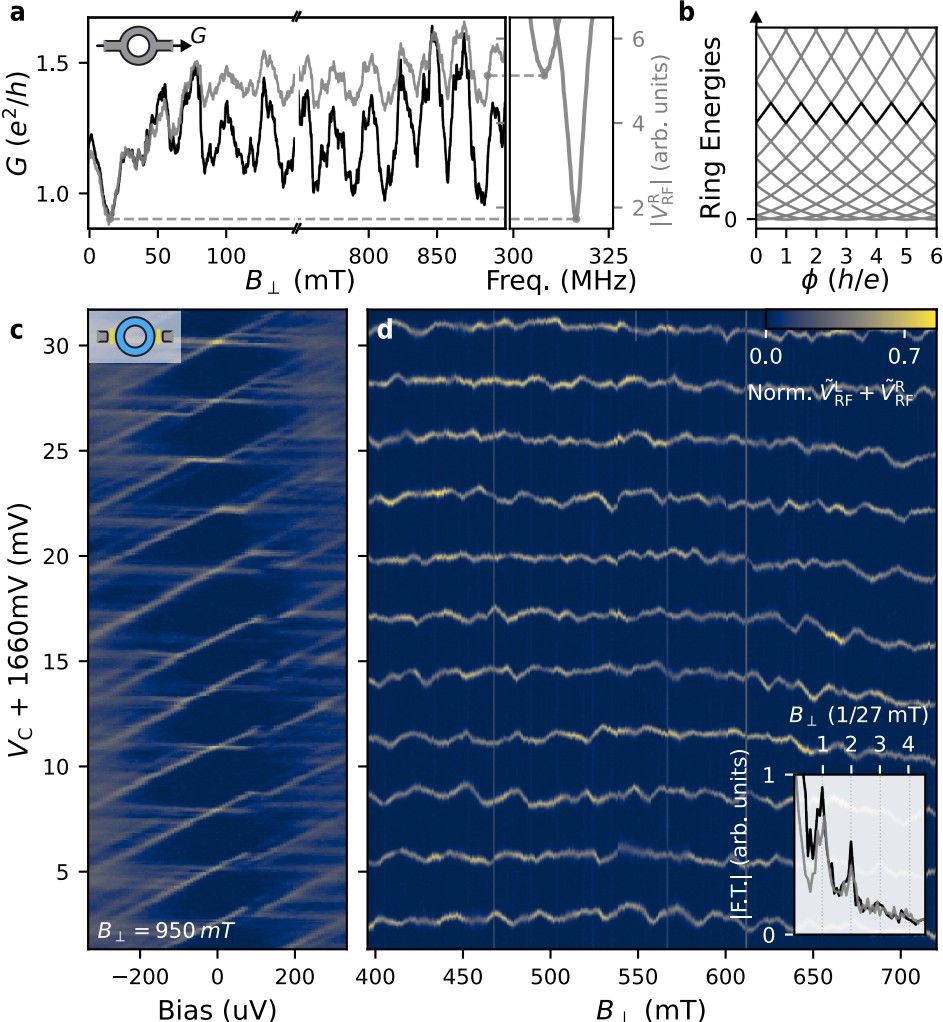

Figure 2: Phase-coherent transport and extended electron states. **(a)** AB oscillations in the open loop configuration depicted in the inset. Measurements are at zero bias voltage, of 4-terminal lock-in conductance (black) and of the absolute reflected signal (gray) from the resonator coupled to the $V_R$ electrode. Here, $|V_{RF}^R|$ is taken at the field-dependent resonance amplitude minimum (right). On the right, example frequency sweeps from which the minimum signal is calculated are shown. $h/e$ and $h/2e$-periodic oscillations are visible in both the conductance and in the RF signal. **(b)** Single-particle energies for a thin ring $\propto (e\phi/h + l)^2$ for $l \in \mathbb{Z}$ where $\phi$ is magnetic flux. The tenth lowest energy state is highlighted, showing that energies for fixed electron number oscillate in a zig-zag fashion. **(c)** Coulomb diamonds with the device configured into a ring-shaped QD (depicted in the inset) at $B_\perp = 950$ mT. The sum of normalized signals from both gate resonators is plotted, centered about the signal in Coulomb blockade. A consistently finite excitation energy is visible. **(d)** Zero-bias Coulomb resonances as a function of $B_\perp$, with measurement frequencies adjusted to be near resonance at each $B_\perp$ value. *Inset:* Normalized absolute Fourier transform of the resonance $V_C$ position (black) and signal height (gray) averaged across all Coulomb resonances. Both have clear peaks at an $h/e$ periodicity of 27 mT.

# 4 Flux-tunable interdot coupling

Having established phase coherence of the 2DEG loop, we next consider the case of a loop comprising two quantum dots threaded by a magnetic flux, illustrated in Fig. 3(a). For this system, we expect magnetic flux to tune the effective interdot tunnel coupling. This is in contrast to studies embedding QDs into semiconducting rings where one trajectory involving tunneling through a QD could interfere with trajectories involving the other loop arm, potentially containing a second QD [48–57]. Assuming that at each interdot charge transition both QDs are described by a single fermionic level, the DQD can be represented as a two-level system with an effective coupling matrix element $t_{\text{eff}} \equiv t_{\text{T}} + t_{\text{B}}$. Here, we define $t_{\text{T}}$ and $t_{\text{B}}$ as the interdot coupling due to the top and bottom arms, respectively. Under the Peierls substitution, a magnetic flux $\phi(B_{\perp})$ imparts a phase on each coupling [7]. Using an appropriate choice of gauge, we then have

$$|t_{\text{eff}}| = \sqrt{|t_{\text{T}}|^2 + |t_{\text{B}}|^2 + 2|t_{\text{T}}t_{\text{B}}| \cos(2\pi\phi/\phi_0)}, \tag{1}$$

assuming $t_{\text{T}}$ and $t_{\text{B}}$ had equal phases at zero field. Via quantum capacitance, $t_{\text{eff}}(\phi)$ imparts a frequency shift on QDL's gate resonator with a maximal magnitude in the ground state which is proportional to $1/|t_{\text{eff}}|$ at the charge degeneracy point [31, 33]. Consequently, we expect the frequency shift to oscillate periodically with $\phi$. In Figs. 3(b,c), we plot the expected dependence of the resulting frequency shift on flux [31, 33].

Experimentally, we realize this system as a loop-shaped DQD with chemical potentials tuned by voltages $V_{\text{L}}$ and $V_{\text{R}}$. To focus on interdot transitions where the signal contains information about the interdot tunnel coupling $t_{\text{eff}}$, we lower $V_{\text{BS}}$ and $V_{\text{BD}}$ until tunneling rates to the leads are immeasurably small, but still nonzero so that the system can reach its ground charge state. Meanwhile, we form the DQD by lowering $V_{\text{BT}}$ and $V_{\text{BB}}$ into a regime of moderate tunneling, such that interdot transitions exhibit a substantial quantum capacitance signal. The barriers are tuned to be approximately equal based on DC current measurements (Appendix C).Coulomb diamond measurements demonstrate a varying but finite level spacing above 70 μeV in both QDs (Appendix D) [58], such that the DQD is well-described by two coupled but potentially spin-degenerate fermionic levels [59]. Maintaining a finite excitation energy on both QDs despite their large lithographic size is achievable due to the low effective mass of roughly $0.016m_{\text{e}}$ in the 2DEG [40], which favors confinement.

Selecting a single interdot transition in this regime, we measure gate and frequency dependent traces of the gate L resonator's response $V_{\text{RF}}^{\text{L}}$ as a function of $B_{\perp}$, aiming to extract $|t_{\text{eff}}|$. At each point in the gate space, we fit the results to an asymmetric resonator model to extract the resonance frequency shift $\Delta f_0^{\text{L}}$ [60–62]. As no resonator losses were measured over this interdot transition, the resonator response may be described as a quantum capacitance $C_{\text{q}}$ loading the bare capacitance $C$ and inductance $L$ of the resonance frequency as $f_0 = 1/2\pi\sqrt{L(C + C_{\text{q}})}$. Accordingly, we fit the $V_{\text{L}}$ dependence of $\Delta f_0^{\text{L}}(C_{\text{q}})$ to a thermal quantum capacitance model described by

$$C_{\text{q}} = 2(e\alpha_{\text{L}})^2 \frac{|t_{\text{eff}}|^2}{(\Delta E)^3} \tanh\left(\frac{\Delta E}{2k_{\text{B}}T}\right), \tag{2}$$

to extract $|t_{\text{eff}}|$, where

$$\Delta E \equiv \sqrt{\alpha_{\text{L}}^2(V_{\text{L}} + V_{\text{L}}^{\text{off}})^2 + 4|t_{\text{eff}}|^2}, \tag{3}$$

is the energy splitting between the two dot levels involved in tunneling [31, 33]. The lever arm $\alpha_{\text{L}} = 0.18$ and electron temperature $T = 71\,\text{mK}$ are optimized simultaneously for all field values to produce the minimal fit error (Appendix E). Subsequently they are fixed, with the only other free parameters being the center offset $V_{\text{L}}^{\text{off}}$ of the transition and $f_0$ in the Coulomb blockade.

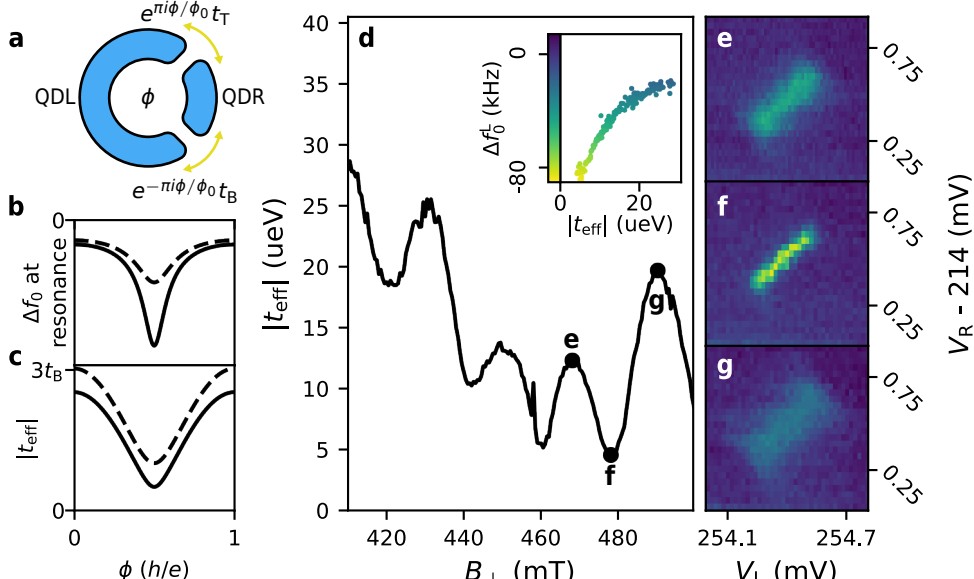

Figure 3: Tuning DQD hybridization with flux. **(a)** Diagram of a DQD ring threaded by a magnetic flux $\phi(B_\perp)$. **(b),(c)** Schematic mapping of $|t_{\text{eff}}|$ as a function of magnetic flux $\phi$ **(c)** into a final resonator frequency shift $\Delta f_0(\phi)$ at charge resonance **(b)**, shown for $t_T = 1.5 t_B$ (solid) and $2t_B$ (dashed). For sizable $|t_{\text{eff}}|$ the frequency shift is $\propto 1/|t_{\text{eff}}|$ [31, 33]. **(d)** Fit $|t_{\text{eff}}|$ values from the frequency response of the gate L resonator as a function of $B_\perp$ for a single interdot transition. The tunnel coupling oscillates periodically with varying contrast and amplitude. The inset defines the charge stability diagram (CSD) color scale and plots the approximately $\propto 1/|t_{\text{eff}}|$ correspondence between the fit $|t_{\text{eff}}|$ and maximum observed $\Delta f_0^L$ for each $B_\perp$ in **(d)**. **(e-g)** Select CSDs at the $B_\perp$ values labeled in **(d)** showing the line shape of $\Delta f_0^L$ across the interdot transitions for different tunnel couplings.

The resulting values of $|t_{\text{eff}}|$ are plotted in Fig. 3(d), where oscillations of $|t_{\text{eff}}|$ are clearly visible. In Figs. 3(e-g), we show examples of frequency shifts of the gate L resonator for several values of $B_\perp$, where we see that for smaller tunnel couplings the transition appears to be more narrow, but with a stronger frequency shift. In particular, the tunnel coupling in general does not reach zero at its minima, suggesting that $t_T$ and $t_B$ are not precisely equal, as exemplified in Fig. 3(c). The average value of $|t_{\text{eff}}|$ between oscillations also varies unpredictably, indicating that the wave functions of the involved states change over the range of multiple flux periods. Nevertheless, with this measurement we explicitly demonstrate control of the hybridization between two fermionic levels with magnetic flux.

# 5 Limits of flux-tuned tunnel coupling readout

For applications to topological qubits using QDs potentially containing many electrons, one must choose a particular dot level to optimize tunnel coupling readout. Therefore, in the same DQD regime as in Sec. 4, we proceed to study the variance of the oscillation amplitude in a broader field range and for multiple transitions, focusing on the 16 transitions shown in Fig. 4(a). There, similar to measurements of the ring-shaped QD, we plot the absolute deviation of the complex reflection signal of QDL's resonator from its average value in Coulomb blockade: $\tilde{V}_{\text{RF}}^L$. The complex signal is a one-to-one function of the frequency shift of QDL's resonator and is inversely proportional to $|t_{\text{eff}}|$ for substantial $|t_{\text{eff}}|$ [63]. An even-odd alter-

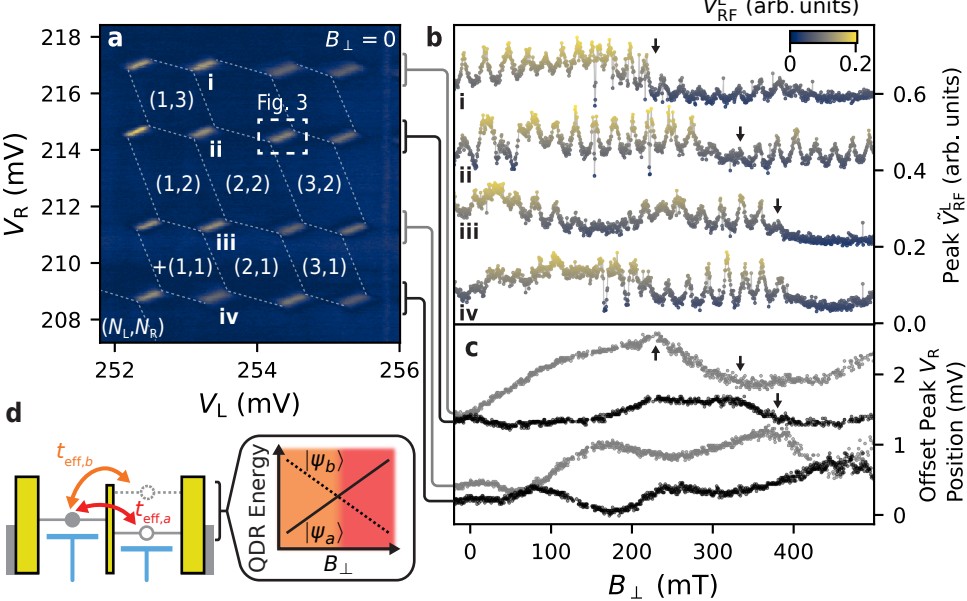

Figure 4: Flux-tunable hybridization of the DQD across multiple dot levels. **(a)** CSD with no applied field showing the window of 16 interdot transitions probed over a sweep of $B_\perp$. Dashed lines show the approximate boundaries of stable charge regions, because weak coupling of the QDs to the leads makes only interdot transitions visible in the gate L resonator's signal. Several charge regions are labeled with their relative charge states up to an offset $(N_L, N_R)$ for unknown even reference charges $N_L$ and $N_R$ on QDL and QDR, respectively. **(b)** Peak signal deviation from Coulomb blockade $\tilde{V}_{RF}^L$ of the four numeral-labeled transitions as a function of $B_\perp$, offset by 0.18 arb. units. **(c)** Peak positions of interdot transitions in $V_R$ coordinates relative to the lowest peak, averaged across all four columns of transitions shown in **(a)**, and offset by 2.32 mV. The offset voltages vary linearly with the addition energies of QDR, so that anticrossings in the positions correspond to anticrossings between electron states of QDR. The black arrows show example points where a correlation can be observed between the oscillation amplitude of $\tilde{V}_{RF}^L$ and anticrossings of QDR states. **(d)** Schematic describing the kinks in **(c)** and sudden changes in the $|t_{eff}|$ oscillations of **(b)**. If a state $|\psi_b\rangle$ overtakes another state $|\psi_a\rangle$ as the ground state of QDR, and the former has a different tunnel coupling to the ground state of QDL, then a sudden change in $|t_{eff}|$ and its oscillation amplitude may occur at this crossing.

nation in the transition spacing both along the $V_L$ and $V_R$ axes suggests that both QDs have spin degenerate levels with a finite level spacing in this window. We sweep $B_\perp$, measuring new CSDs of the 16 transitions at a single measurement frequency adjusted to remain close to resonance. From these CSDs, we extract the maximum $\tilde{V}_{RF}^L$ signal and the approximate peak position in the gate space for all transitions.

We plot in Fig. 4(b) the peak signal height—proportional to $1/|t_{eff}|$ except when $|t_{eff}|$ is very small—for the column of transitions enumerated in Fig. 4(a). For all four transitions, $h/e$-periodic oscillations of the peak height are clearly seen in some ranges of $B_\perp$. There, we identify four distinct features. First, some regions in Fig. 4(b) present visible oscillations in a relatively small signal. One such region appears between $B_\perp = 220$ and $B_\perp = 400$ mT for Transition i. As schematically depicted in Fig. 3(b,c), this corresponds to large average $|t_{eff}|$ and asymmetric barriers. Large tunnel couplings lead to a small frequency shift while asymmetry reduces the amplitude of the oscillations. Second, for smaller mean values of $|t_{eff}|$ the signal

variation with flux is much greater since $|\mathrm{d}\Delta f_0^{\mathrm{L}}/\mathrm{d}|t_{\mathrm{eff}}||$ is larger, as seen for transition iv in the range 280 mT to 400 mT for example. Third, Transition iv at low fields exhibits a substantial peak height, indicating a small tunnel coupling, but a very weak oscillation contrast. This suggests that the tunnel barriers are tuned by $B_\perp$ to be substantially asymmetric in this field range. Finally, a sudden drop of the peak height to near zero sometimes appears near the oscillation maximum for transitions i and ii. We expect this to be a result of $|t_{\mathrm{eff}}|$ being small enough near the maximum peak height that thermal excitations and Landau-Zener transitions populate the excited DQD state, suppressing quantum capacitance (see Appendix G for a more detailed argument) [64,65]. Importantly, this also suggests that $|t_{\mathrm{B}}| \approx |t_{\mathrm{T}}|$ in those cases.

Differences between these scenarios are known to have consequences when sensing tunnel coupling to manipulate or measure qubits [34,66,67]. Probing the tunnel coupling with gate sensing in the regime of very weak tunneling gives a sharp change in the resonator signal for small changes in $|t_{\mathrm{eff}}|$, allowing one to couple QDs weakly to the qubit of interest. Conversely, the signal is also sensitive to small changes in flux in this case. Certain topological qubit proposals also rely on a substantial tunneling magnitude for their operation [8].

To better understand the results of Fig. 4(b), we now consider the influence of the specific electronic levels involved on the amplitude of the tunnel coupling oscillations. To this end, we plot the relative position $V_{\mathrm{R}}$ of interdot transitions averaged across all four columns in Fig. 4(c) and offset by the inferred product of their charging energy and gate lever arm: 2.32 meV. This position is proportional to the excitation energies of the different QDR levels [68,69], and we observe that they are nearly spin-degenerate at zero field. Kinks can be seen in the peak positions, indicating (anti-)crossings between levels of QDR, depicted schematically in Fig. 4(d). At several fields, with examples highlighted by black arrows in Fig. 4(b,c), sudden changes in the average peak height and oscillation contrast of a transition appear correlated with anticrossings of QDR levels. We hypothesize that variation in wave function overlap of different levels with field, as well as the particular levels involved, can have a drastic effect on $t_{\mathrm{T/B}}$. As the cartoon in Fig. 4(d) illustrates, it may be the case that two different states of QDR have different wave-function overlaps with the ground state of QDL, and vice-versa. In particular, transitions between states of opposing spin have $t_{\mathrm{eff}}$ determined by spin-orbit coupling strength [26,70,71], while transitions between states of the same spin do not. Given the large out-of-plane $g$-factor of these 2DEGs [40], it was difficult to independently study spin and flux effects. Additionally, some changes in the mean peak height and oscillation contrast have no obvious correlation with QDR excitation energies, but we note that changes in the ground state of QDL as a function of field also affect $t_{\mathrm{eff}}$. Hence, for any application requiring hybridization readout between QD levels, the specific levels used must be optimized for a given magnetic field range.

Lastly, we compare the differences in tunnel coupling readout contrast for regimes of different $V_{\mathrm{BT/BB}}$ and thus average $t_{\mathrm{T/B}}$ values. From Eq. 1 we expect that for nearly equal $t_{\mathrm{B}}$ and $t_{\mathrm{T}}$, large tunnel couplings should produce the best oscillation contrast, since the tunnel coupling ranges from $|t_{\mathrm{T}}| + |t_{\mathrm{B}}|$ to nearly zero. We therefore conduct measurements analogous to those in the intermediate coupling regime of Fig. 4 for other coupling regimes, with results summarized in Fig. 5 and shown in more detail in Appendix H. Namely, we first bin the peak heights for a given regime into windows equal to the $h/e$ periodicity extracted from their average Fourier transform (Fig. 5(d)). Next, we plot bars spanning the minimum $\tilde{V}_{\mathrm{RF}}^{\mathrm{L}}$ peak height to the maximum for whichever of the 16 transitions maximizes this difference in a given field bin. In addition to the dataset from Fig. 4, datasets for more negative (closed) and less negative (open) barrier gate voltages are shown in blue and green, respectively. As a control, in orange we show the data for an in-plane field sweep over the same transitions considered in Fig. 4, where no oscillations are seen. Compared to the red 'intermediate' coupling regime, the more closed-off regime shows on average a larger variation in peak height

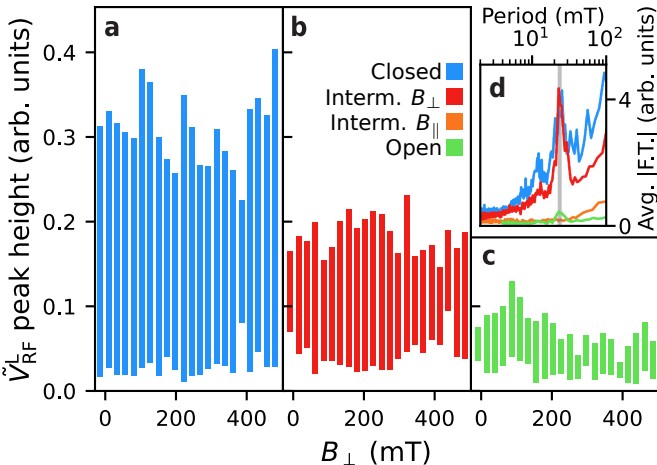

Figure 5: Contrast of DQD Tunnel Coupling Variation in Different Regimes. **(a-c)** Bars showing maximal peak height variation on a single interdot transition spanning the distance between the smallest and largest observed $\tilde{V}^{\mathrm{L}}_{\mathrm{RF}}$ peak height, binned within one $h/e$ period of 23.5 mT and plotted for three different regimes of tunnel barrier tuning. Of the 16 interdot transitions tracked in each dataset, only the bar for the transition with the largest signal variation for each period is shown. **(a)** summarizes a $B_\perp$ sweep in a regime of weak interdot tunneling with more negative barrier voltages, while **(c)** shows data for strong tunneling and less negative barrier voltages. **(b)** corresponds to the intermediate tunnel barrier data from Fig. 4. The largest contrast in the signal generally occurs within the weak coupling regime. **(d)** Absolute Fourier transforms in each regime averaged across all 16 transitions. Orange represents a sweep of the in-plane field for the same transitions and tuning as the intermediate regime. A vertical line shows the peak at 23.5 mT.

across a single $h/e$ period, due to the increased slope of $\Delta f_0^{\mathrm{L}}$ with flux as described above. The open regime shows very weak oscillation contrast despite the tunnel barriers exhibiting similar resistances to each other (Appendix C), suggesting that larger coupling regimes are more sensitive to slight asymmetries between $t_{\mathrm{T}}$ and $t_{\mathrm{B}}$. If the percent difference between $|t_{\mathrm{T}}|$ and $|t_{\mathrm{B}}|$ is non-negligible, then the maximum flux-tuned difference in quantum capacitance signals (proportional to $1/||t_{\mathrm{T}}| - |t_{\mathrm{B}}|| - 1/(|t_{\mathrm{T}}| + |t_{\mathrm{B}}|)$ for real $t_{\mathrm{T/B}}$) becomes smaller for larger average tunnel couplings. Consequently, for flux-tuned qubit readout and manipulation schemes where the state is encoded in the sum or difference of two tunnel couplings [10–12, 15], the optimal readout fidelity may occur for weak overall couplings.

# 6  Conclusions & outlook

Herein we have measured a tunable hybridization between two electronic levels threaded by a magnetic flux for the first time. Using gate-based RF reflectometry implemented in a phase-coherent InSb$_{0.86}$As$_{0.14}$ 2DEG, we measured $h/e$-periodic oscillations of tunnel coupling between the levels of two QDs arranged in a loop. Even for nearly symmetrically tuned interdot tunnel barriers, the coupling was not generically suppressed at its minima, exhibiting a high degree of variability in magnitude and contrast of the tunnel coupling oscillations. We inferred that this variability is in part dependent on the specific QD levels involved. Finally, we found that, given the inherent difficulty of symmetrically tuning two tunnel barriers in parallel, the best signal contrast across an oscillation period occurs for relatively weak average interdot

tunnel couplings [34]. On the other hand, tuning a tunnel barrier strength as a function of flux while probing the gate reflectometry signal at an interdot charge resonance serves in itself as a method for tuning $|t_T|$ and $|t_B|$ to be equal. In this approach one would exploit the fact that $|t_{eff}|$ has a minimum of $|t_T - t_B|$, and target the barrier strength where Landau-Zener transitions suddenly suppress the signal near its maximum as a function of flux, as described in Appendix G. This work establishes a prerequisite for the readout of qubits formed in topological nanowires and Kitaev chains [10–12,16,72]. It also demonstrates a new mechanism by which the effective coupling between localized electronic states can be tuned and illustrates its limitations, applicable to semiconducting spin and charge qubits [22–24]. Even when undesirable, flux-tuned tunnel couplings may arise in two-dimensional QD arrays [5], as direct tunneling or cotunneling between QDs can occur via more than one trajectory in this case.

# Acknowledgments

The authors are grateful to J.V. Koski, L.P. Kouwenhoven, and F. Borsoi for helpful discussions and input on the manuscript and to L.P. Kouwenhoven for initiating the project.

**Funding information** The authors also acknowledge financial support from Microsoft Quantum and the Dutch Research Council (NWO). F.K.M. acknowledges support from NWO under a Veni grant (VI.Veni.202.034).

**Data availability** Raw data, analysis code, and scripts for plotting the figures in this publication are available from Zenodo [73].

**Author contributions** C.G.P. and I.K. fabricated the device using a 2DEG heterostructure provided by D.X., C.T., and M.J.M.. C.G.P. and M.C. conducted the measurements with input from L.H. and F.K.M. F.K.M. and S.G. supervised the project. C.G.P. analyzed the data and wrote the manuscript with input from all authors.

# A  Device design & fabrication

Here we describe in more detail the design considerations in fabricating the measured device. One equivalent in design to the one measured from the same chip is shown in Fig. 6(a). Initially, the chip is covered with a <10 nm epitaxial layer of Al which was selectively etched away everywhere except in a region to the left and right of the pictured device to form leads, exposing the InSb$_{0.86}$As$_{0.14}$ 2DEG heterostructure. The 2DEG itself—where electrons conduct—is formed near the surface of the heterostructure. Details of the heterostructure can be found in Ref. [40]. Next, the 2DEG was etched away except in a region close to the active device and along a roughly 140 μm path connecting it to the Al leads, forming a mesa. The fact that the Al leads are superconducting and separated by roughly 6.3 μm of conducting 2DEG from the active device means that four-terminal measurements of the device conductance are possible, including a small resistive contribution from the exposed 2DEG portion of the leads. To do so, we simply bond two DC lines each to the superconducting source and drain leads. We then alternated between using atomic layer deposition to deposit roughly 20 nm Al$_2$O$_3$ dielectric layers then evaporating Ti/Pd gate layers to form three electrically isolated gate layers. Each layer also contains thicker coarse gate leads (not shown), required to facilitate climbing the mesa. The 2DEG mesa on which the device was fabricated conducts, so forming a loop required application of negative voltages both along the outer perimeter of the loop, as well as

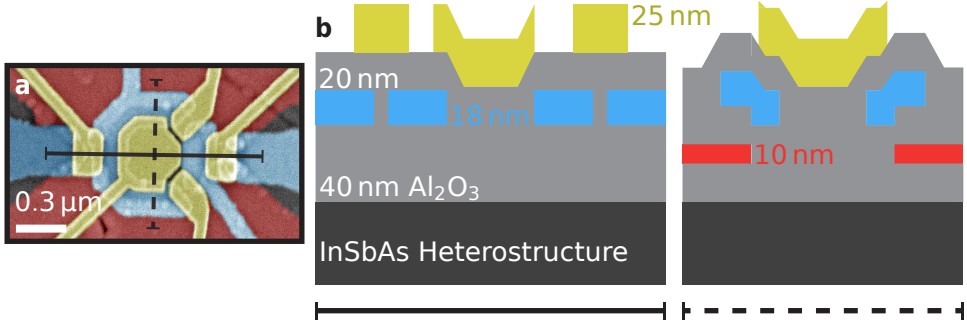

Figure 6: Device design and layer stack. **(a)** False-color scanning electron micrograph for a device nominally equivalent to the one measured on the same chip. The colors are encoded by gate layer, of which there are three, instead of by the gates' purpose as was done in Fig. 1(a). **(b)** Cross-sections approximately depicting the layer stack of the device along the solid and dashed lines shown in **(a)**. Thicknesses of the dielectric and Ti/Pd gate layers are relatively to scale, but the widths are not, and the topography is only schematically depicted.

in the hole in the center. Fabricating a DQD in this loop further necessitated plunger gates to tune the chemical potential of the QDs and gates to form barriers between them and to the contacts. One option to satisfy these requirements is to pattern depletion gates in a layer above the plunger gates needed to tune the QDs, however in this case the leads of the lower layer gates were found in previously measured devices to screen the depletion gate voltage and prevent forming a stable loop. Hence, it was topologically required to fabricate three gate layers in order to both have an outer depletion gate underneath the plunger and barrier gates, as well as a central depletion gate which can cross over the plunger gates to deplete the center of the loop. The corresponding layer stack is schematized in Fig. 6(b), with details of the 2DEG heterostructure underneath given in Ref. [40]. A third gate layer had the added advantage that tunnel barriers could be made effectively more narrow, since barrier gates in the third layer may overlap with plunger gates in the second layer. Notably, thin wires with very high resistance were also fabricated on-chip in series with the lower depletion gate leads, such that cross-capacitances between gates used for RF reflectometry would not shunt the resonator signal through lossy DC lines to ground.

## B Calculating the RF signal deviation from Coulomb blockade

The scattering parameters $V_{RF}^L$ and $V_{RF}^R$ measured in the reflectometry circuit are complex and at Coulomb resonance the signal information is stored in both their real and imaginary components. To illustrate this, we plot the histogram of measured $V_{RF}^L$ values using the dataset of Fig. 4(a) in Fig. 7. A large concentration of points is centered around the Coulomb blockade signal (denoted $V_{RF}^{L0}$) away from $V_{RF}^L = 0$, while an elongated distribution of points corresponds to the signal around a Coulomb resonance. The vector between these two groupings of measured values, illustrated with an arrow for an arbitrary $V_{RF}^L$ on Coulomb resonance, contains most of the signal information. Hence, to plot a real quantity representing the RF signal while excluding the minimum possible amount of information, we plot the magnitude of this vector, denoted $\tilde{V}_{RF}^L \equiv |V_{RF}^L - V_{RF}^{L0}|$. We note that a second elongated distribution of points appears in Fig. 7 oriented horizontally. This arises from a stray charge resonance unrelated to the QDs but sensed by resonator L [74], appearing as a vertical resonance along the right side of Fig. 4(a).

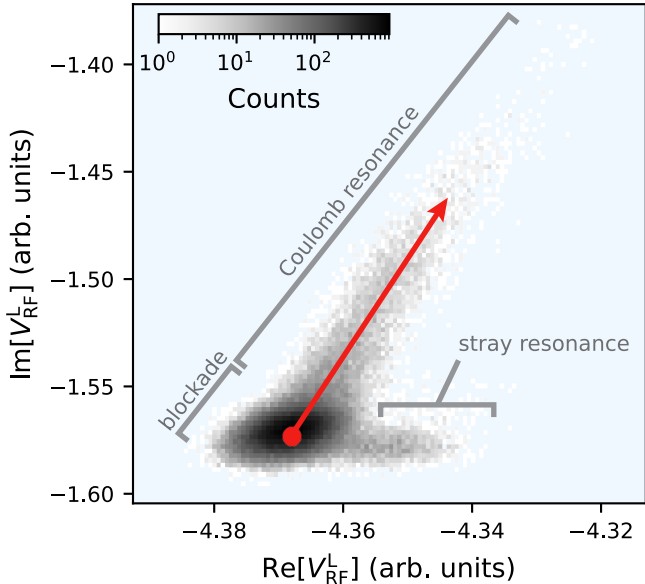

Figure 7: Histogram of the measured complex $V_{\rm RF}^{\rm L}$ values from the dataset of Fig. 4. The extracted value of $V_{\rm RF}^{\rm L0}$ for this dataset is plotted in red, and is roughly centered over the clustering of points corresponding to the Coulomb blockade signal. $\tilde{V}_{\rm RF}^{\rm L}$ is calculated as the absolute deviation of the signal from this point.

To estimate $V_{\rm RF}^{\rm L0}$, we use two different methods. For data shown in Fig. 2, we take the mean $V_{\rm RF}^{\rm L}$ over a rectangular window observed to correspond to Coulomb blockade from an initial inspection of $|V_{\rm RF}^{\rm L}|$ as $V_{\rm RF}^{\rm L0}$. This technique is robust provided that charge jumps do not move Coulomb resonances into the window. For the data shown in Figs 4, 5 and 11, however, we use a modified median of the data since it can be automatically calculated without specifying a window corresponding to Coulomb blockade. Namely, we first extract the lowest 50 % of $V_{\rm L}$ rows in the dataset in terms of their $V_{\rm RF}^{\rm L}$ standard deviation. This is because rows with high standard deviation are expected to contain Coulomb resonances since the signal varies more from its Coulomb blockade value. From this subset of points, we take the median as $V_{\rm RF}^{\rm L0}$. To illustrate this, we plot the $V_{\rm RF}^{\rm L0}$ value extracted with this method in Fig. 7 as a red point. We see that it is roughly centered over the clustering of points corresponding to Coulomb blockade. Note that the same steps are used for $V_{\rm RF}^{\rm R}$ data as used in Figs. 2(b) and 2(c). A different Coulomb blockade value is taken at each magnetic field value in the case of a field sweep, since the field affects the resonator's line shape and resonance frequency.

## C Tuning symmetric parallel tunnel barriers

To tune the bare tunneling strengths $t_{\rm T}$ and $t_{\rm B}$ to be approximately equal, we select voltages on their corresponding barrier gates such that each admits the same instantaneous conductance when the other barrier is completely closed off. This procedure is summarized in Fig. 8. For this method to be valid, we must assume that the barrier gates have a negligible capacitive cross coupling, as evidenced by the approximate rectangular shape of their two-dimensional pinch-off map shown in Fig. 8(a).

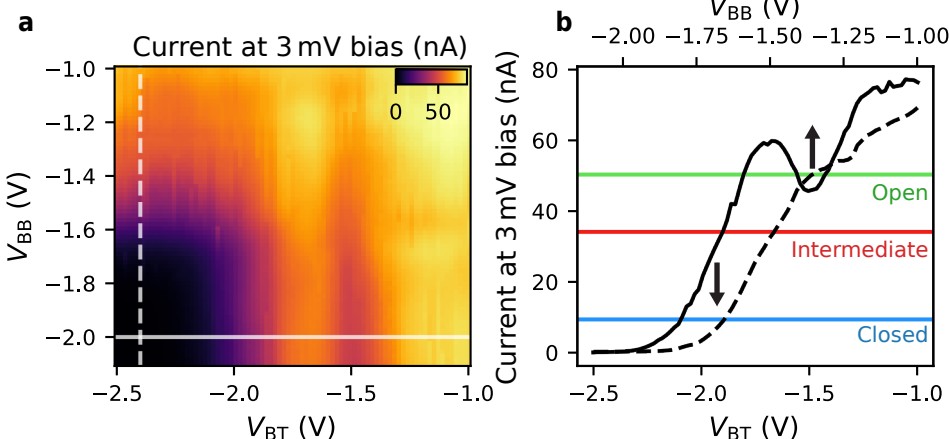

Figure 8: Pinch-off scans for approximately symmetric barrier tuning. **(a)** Current through the device at 3 mV applied bias voltage as a function of $V_{\mathrm{BT}}$ and $V_{\mathrm{BB}}$, tuned into an otherwise open loop. The roughly rectangular shape of the zero-current region implies a weak cross-coupling between gates BT and BB. Linecuts where BT or BB are closed (white lines) can thus be used to select barrier voltages for roughly equal resistance. **(b)** Linecuts from the current map of **(a)**. To tune for the intermediate coupling regime of Fig. 4 (red), or the more closed off (blue) and open (green) regimes described in Fig. 5, $V_{\mathrm{BT}}$ and $V_{\mathrm{BB}}$ voltages are chosen such that when the opposite barrier is pinched off, they both admit roughly the same current. The relatively large bias reduces the influence of QD states under the barriers on the measurement. The instantaneous conductance through the parallel barriers in the closed, intermediate, and open voltage regimes are roughly 0.1, 0.3, and 0.4 $e^2/h$.

## D Coulomb diamonds

Coulomb diamonds of QDL and QDR are shown in Fig. 9, from which we observe that both exhibit a consistently finite excitation energy between electronic levels.

## E Fitting procedure for extraction of tunnel couplings

Herein we detail the procedure used to extract the effective tunnel coupling magnitude of a DQD ($|t_{\mathrm{eff}}|$ in the main text), given a CSD spanning an inter-dot charge transition with a frequency-dependent response measured at each point for a resonator coupled to one of the QD's gates. The parametric capacitance for a gate at voltage $V_{\mathrm{g}}$ primarily coupled to a single charge island or QD (indexed by $i$) out of multiple potentially coupled islands is

$$C_{\mathrm{p}} = \tilde{\alpha}_i |e| \frac{\mathrm{d}\langle \hat{n}_i \rangle}{\mathrm{d}V_{\mathrm{g}}} \,, \tag{E.1}$$

where $\langle \hat{n}_i \rangle$ is the expectation value of charge on QD $i$ and $\tilde{\alpha}_i$ is a lever arm of the gate's coupling to the quantum modified by mutual capacitances of this QD to other charge islands in the system, see Appendix F for further details. In essence, the large inter-dot capacitance of the system when tuned into the DQD regime (as can be inferred from the inter-dot transition width in gate space relative to the spacing between transitions in Fig. 4(a) [69]) lowers the effective lever arm of the gate to the sensed QD. Consequently, we must fit for $\tilde{\alpha}_i$ independently, since it is not expected to agree with the lever arms extractable from the Coulomb diamond

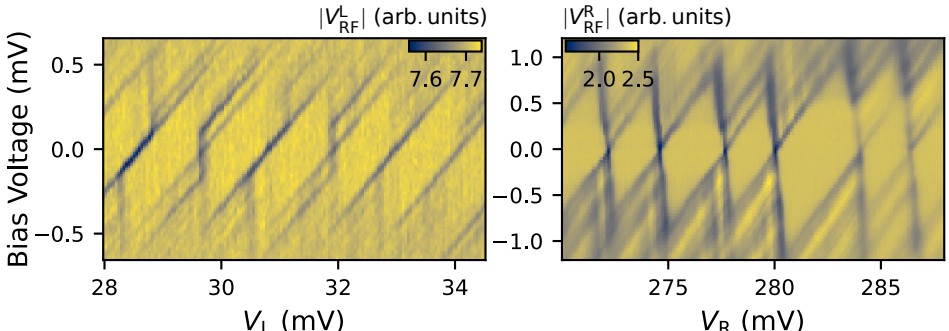

Figure 9: Coulomb diamonds of QDL **(a)** and QDR **(b)**. The single QDs are tuned such that both the relevant lead barrier as well as $V_{\text{BT}}$ and $V_{\text{BB}}$ are in a weak tunneling regime. Magnitude of the reflectometry signal near the resonance frequency of their respective plunger gates' resonators is plotted. A varying but finite level energy spacing is visible for both QDs larger than the linewidth.

measurements of Fig. 9. This parametric capacitance can be calculated from the fitted resonator frequency $f_0$ as $C_{\text{p}} = 1/4\pi^2 L f_0^2 - C$ where $L$ and $C$ are the resonator's bare inductance and capacitance, respectively. In practice, we approximate $L$ at zero magnetic field as its simulated value for the resonator's inductor coil. We calculate $C$ from the resonance frequency in Coulomb blockade, where $C_{\text{p}}$ is assumed zero. At each value of the out-of-plane magnetic field $B_\perp$, we assume that in Coulomb blockade the only shift in the resonator frequency is due to changes in $L$, such that from frequency fits at each field we can extract $L(B_\perp)$ assuming $C(B_\perp)$ is fixed. Thus, the parameters $L$ and $C$ are fixed by measurements and not varied in the subsequent fits described below.

As an explicit model for parametric capacitance, we consider the model of Refs. [31, 33] for a DQD coupled to a phonon bath. Near an inter-dot transition, this model considers two charge states with an excess electron residing either on a discrete fermionic mode of the sensed QD, or a mode of a second QD. These two modes are coupled by tunnel coupling $t_{\text{eff}}$, and the detuning between their energies is given by $\varepsilon = \tilde{\alpha}_i(V_g - V_g^{\text{off}})$ where the offset $V_g^{\text{off}}$ determines the transition position in gate space. In this model, the parametric capacitance is found to be

$$C_{\text{p}} = \underbrace{2(e\tilde{\alpha}_i)^2 \frac{|t_{\text{eff}}|^2}{(\Delta E)^3} \tanh\left(\frac{\Delta E}{2k_B T}\right)}_{\equiv C_{\text{q}}(\varepsilon)} + \frac{(e\tilde{\alpha}_i)^2}{4k_B T}\left(\frac{\varepsilon}{\Delta E}\right)^2 \frac{\gamma^2}{\omega^2 + \gamma^2}\cosh^{-2}\left(\frac{\Delta E}{2k_B T}\right), \qquad \text{(E.2)}$$

where $\Delta E \equiv \sqrt{\varepsilon^2 + 4|t_{\text{eff}}|^2}$ is the energy splitting of the charge qubit and $\omega$ is the angular resonator measurement frequency. The first term above corresponds to quantum capacitance while the second corresponds to so-called tunneling capacitance. The parameter $\gamma$ quantifies incoherent tunneling due to phonon absorption and emission, and in principle is another parameter we must include in our fit of $C_{\text{p}}$ to extract $|t_{\text{eff}}|$.

A resistive contribution to the effective impedance of the sample known as Sisyphus conductance arises, however, whenever there is substantial tunneling capacitance [31, 33], which would lower the resonator internal quality factor near the transition. In our fits of the frequency-dependent CSDs, the change in resonator quality factor was not discernible at the inter-dot transition, indicating that Sisyphus resistance and likely tunneling capacitance can be neglected in our fits. This also indicates that all information about inter-dot tunneling is contained in the frequency shift $\Delta f_0$, such that we may solely fit $\Delta f_0(V_g)$ to extract $|t_{\text{eff}}|$, rather than simultaneously fitting the frequency shift and quality factor. Regardless, in Fig. 10(c) we show that maximizing the contribution of tunneling capacitance leads to a negligible change

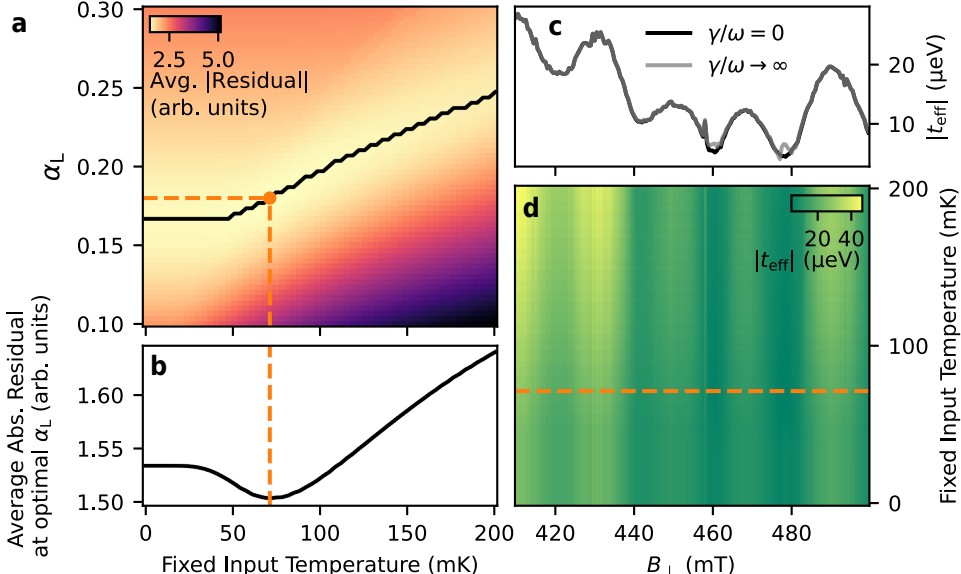

Figure 10: Optimization of tunnel coupling fits. **(a)** The mean absolute residual difference between the fit $C_p$ line shape of the inter-dot charge transition as a function of $V_L$ and the $C_p$ values extracted from fit frequency shifts of QDL's resonator. The black line shows the fixed $\alpha_L$ value minimizing the residual error for each fixed $T$. **(b)** The mean residual error with $\alpha_L$ fixed at its optimal value shown in **(a)** for each fixed value of $T$. A clear minimum is found at $T = 71\,\text{mK}$ and $\alpha_L = 0.18$. **(c)** The extracted $|t_{\text{eff}}|$ for zero tunneling capacitance ($\gamma = 0$) and maximal tunneling capacitance, which saturates as $\gamma \to \infty$. The presence of tunneling capacitance has a negligible effect on $|t_{\text{eff}}|$ except at very small $|t_{\text{eff}}|$. **(d)** Fit $|t_{\text{eff}}|$ with $\alpha_L$ fixed to the value minimizing fit error for each value of fixed temperature.

to the extracted $|t_{\text{eff}}|$ except for very small tunnel couplings. Hence, we neglect tunneling capacitance for the fits of Fig. 3(d).

Under these constraints, we extract a fitted $C_p(V_g)$ from fitted $\Delta f_0^L$ and our knowledge of $L$ and $C$ described above, and fit the result to

$$C_p = C_q(\alpha_L(V_g - V_g^{\text{off}})) + C_{\text{off}}, \tag{E.3}$$

with $C_q$ as defined above and where we denoted $\tilde{\alpha}_i \to \alpha_L$ as the effective QDL lever arm. In fact, we select five rows of the gate voltage near the center of the transition and fit them simultaneously with the same $|t_{\text{eff}}|$, $\alpha_L$, and $T$, but allow for a different $C_{\text{off}}$ and $V_g^{\text{off}}$ for each row. In other words, we fit multiple traces for values of the other QD's gate voltage near the center of the charge transition in the charge stability diagram. The offset $C_{\text{off}}$ accounts for errors in converting from $\Delta f_0$ to $C_p$. These parameters are fitted independently for each row.

Since $T$ and $\alpha_L$ should be roughly the same at all fields, we sweep different fixed values of these parameters iteratively and choose the values which lead to a minimum total residual across all magnetic field values. We found a global optimum of $T = 71\,\text{mK}$ and $\alpha_L = 0.18$ which minimized the mean absolute fit residual error, see Fig. 10(a,b). This temperature is larger than the roughly $20\,\text{mK}$ temperature of the dilution refrigerator used, which is not unexpected since electron temperature may be raised by connection to higher temperature cables and electronics [58]. Lastly, in Fig. 10(d), we observe that the oscillation amplitude of $|t_{\text{eff}}|$ does vary with increasing temperature used in the fits (with $\alpha_L$ fixed at the optimum shown in Fig. 10(a)), but the oscillations of $|t_{\text{eff}}|$ are consistently present with a period of one flux quantum.

# F  Capacitance formula including mutual capacitances

In order to determine the degree to which mutual capacitances between QDs suppress parametric capacitance, we follow the approach of Refs. [31, 33] to derive an expression for parametric capacitance, additionally considering mutual capacitance effects to second order. We consider the case of $N$ charge islands coupled capacitively to a single gate voltage $V_\mathrm{g}$ via capacitances $C_{gi}$ for $i \in \{1, 2, ..., N\}$, with mutual capacitances between the islands of $C_{ij}$ for $i \neq j$, and other capacitive couplings to ground encompassed by an environmental capacitance $C_{ei}$. The latter includes any capacitances to lead reservoirs, for example. We refer to the total capacitance of each island as $C_i \equiv C_{gi} + C_{ei} + \sum_{j \neq i} C_{ij}$. Note that by definition, we have $C_{ij} = C_{ji}$. The total differential capacitance $C_\mathrm{diff}$ as seen by $V_\mathrm{g}$ can then be written as the sum over differential capacitance contributions of each island

$$C_\mathrm{diff} = \sum_{i=1}^{N} \frac{d \langle Q_i \rangle}{dV_\mathrm{g}} = \frac{d \sum_{i=1}^{N} \langle Q_i \rangle}{dV_\mathrm{g}}, \tag{F.1}$$

where $Q_i$ is the total effective charge on the capacitor $C_{gi}$ as seen by $V_\mathrm{g}$ and the angular brackets denote the statistical average of the charge. In general, this average must include thermodynamic, quantum mechanical, and driving effects.

To solve this expression, we write $\langle Q_i \rangle$ in terms of known capacitances and the expectation values $\langle \hat{n}_i \rangle$ of electron number on each island with charge number operator $\hat{n}_i$. First, by definition of the gate capacitances we may write $\langle Q_i \rangle = C_{gi}(V_\mathrm{g} - V_i)$ where $V_i$ is the electrostatic potential on island $i$. On average, we can write the charge expectation value on island $i$ as a sum over all of the voltage induced charges from each capacitor

$$-|e| \langle \hat{n}_i \rangle = C_{gi}(V_i - V_\mathrm{g}) + \sum_{j \neq i} C_{ij}(V_i - V_j) + C_{ei}V_i, \tag{F.2}$$

with $e$ being the electron charge [69]. Solving for $V_i$ and recalling the definition of $C_i$, we find

$$V_i = \frac{1}{C_i} \left( C_{gi}V_\mathrm{g} + \sum_{j \neq i} C_{ij}V_j - |e| \langle \hat{n}_i \rangle \right). \tag{F.3}$$

By substituting this result for each $V_j$ into the original expression for $V_i$, we may recursively generate expressions for $V_i$ to higher and higher orders in the mutual capacitance lever arms $C_{ij}/C_i$. Doing so twice, substituting the result into the definition of $\langle Q_i \rangle$, and using the resulting expression to calculate $C_\mathrm{diff}$, we find

$$C_\mathrm{diff} = C_\mathrm{geo} + C_\mathrm{p} + \mathcal{O}(C_{ij}^3/C_i^3), \tag{F.4}$$

with contributions from a constant geometric capacitance

$$C_\mathrm{geo} \equiv \sum_{i=1}^{N} \alpha_i \left[ C_i - C_{gi} - \sum_{j \neq i} C_{ij} \left( \alpha_j + \sum_{k \neq j} \frac{C_{jk}}{C_j} \alpha_k \right) \right], \tag{F.5}$$

and a $\langle \hat{n}_i \rangle$-dependent parametric capacitance:

$$C_\mathrm{p} \equiv \sum_{i=1}^{N} \left[ \alpha_i + \sum_{j \neq i} \left( \alpha_j \frac{C_{ij}}{C_j} + \sum_{k \neq j} \alpha_k \frac{C_{ij}C_{jk}}{C_i C_k} \right) \right] |e| \frac{d \langle \hat{n}_i \rangle}{dV_\mathrm{g}}, \tag{F.6}$$

where we have defined the bare lever arms $\alpha_i \equiv C_{gi}/C_i$.

Hence, in addition to large mutual capacitances renormalizing a coupled island's lever arm by increasing $C_i$, there is an additional renormalization factor due to mutual capacitances increasing the effective lever arm. The lowest-order of the latter corrections are multiplied by the cross-capacitive lever arms $\alpha_j \ll 1$, however. Note additionally that as $V_g$ tunes the islands near an inter-dot charge transition between islands $i$ and $j$, the transfer of an electron by this tuning implies $\mathrm{d}\langle \hat{n}_i \rangle / \mathrm{d}V_g \approx -\mathrm{d}\langle \hat{n}_j \rangle / \mathrm{d}V_g$ so that cross-capacitances $C_{gj}$ between the gate voltage and islands other than the island it is designed to sense suppresses the parametric capacitance signal at these transitions [31, 33]. From the slope of successive triple points across multiple inter-dot transitions, these cross capacitances are estimated to be negligible in the measured regimes of this experiment. In this limit, where $V_g$ primarily couples to a single island $i$, but the island itself has relatively larger mutual capacitances to the other islands, we discard terms of the order $C_{ij}\alpha_j/C_j$ for $j \neq i$ but preserve terms to second order in $C_{ij}/C_j$ when multiplied by $\alpha_i \gg \alpha_j$, leading to

$$C_p \sim \left( 1 + \sum_{j \neq i} \frac{C_{ij}^2}{C_i^2} \right) \alpha_i |e| \frac{\mathrm{d}\langle \hat{n}_i \rangle}{\mathrm{d}V_g} = \frac{1 + \sum_{j \neq i} C_{ij}^2/C_i^2}{C_{ei} + C_{gi} + \sum_{j \neq i} C_{ij}} C_{gi} |e| \frac{\mathrm{d}\langle \hat{n}_i \rangle}{\mathrm{d}V_g}, \qquad \text{(F.7)}$$

valid in the limits $C_{ij}/C_i, \alpha_j \ll 1$ and $\alpha_j \ll C_{ij}/C_i$ for all $j \neq i$.

# G Quantum capacitance suppression due to Landau-Zener transitions

Landau-Zener transitions (LZTs) make the used capacitance model inapplicable for small values of $|t_{\text{eff}}| \lesssim \sqrt{\hbar \alpha \delta V_{\text{RF}} f_0}$, where $\delta V_{\text{RF}}$ is the resonator's oscillating voltage amplitude, $\alpha$ is its lever arm to the QD, and $f_0$ is the resonator frequency [65]. There LZTs become frequent, biasing the system towards equal occupation of the excited and ground charge states where quantum capacitance is zero [34]. For a DQD with a short decoherence time, and at zero detuning from the charge transition, the probability of a LZT occurring twice in a resonator cycle is $e^{-2|t_{\text{eff}}|^2/\hbar \alpha \delta V_{\text{RF}} f_0}$ [64, 65]. Due to the sinusoidal nature of the oscillating voltage, a LZT occurring twice in a cycle means that the tunneling electron spends an equal amount of time in the excited DQD state as in the ground state. In other words, the population of the excited state is equal to the population of the ground state when this probability is one. Hence, we expect quantum capacitance to be eventually suppressed for small enough $|t_{\text{eff}}|$, since LZTs become more probable as $|t_{\text{eff}}|$ becomes smaller for fixed $\delta V_{\text{RF}}$. Thermal redistribution also becomes important for small $|t_{\text{eff}}|$, further suppressing the frequency shift [31, 33].

# H Field-dependence of peak heights in different coupling regimes

In this section the full datasets from which Fig. 5 was constructed are shown in Fig. 11, including the dataset used in Fig. 4. The four datasets are measured in three different regimes of inter-dot barrier gate voltage strengths, denoted the 'closed', 'intermediate', and 'open' regimes ordered from the strongest to the weakest barrier gate voltages separating QDL and QDR. Though not shown in the figure, in the closed regime at fixed field values, some transitions occasionally exhibited a jitter from row to row in $V_L$-space. This may be due to very weak coupling from the DQD to the leads resulting in electrons tunneling on to the DQD stochastically as the gate is swept, and may result in unphysical additional suppression of the peak height for some fields. Nonetheless, the prominent peak of the Fourier transform of this data at a periodicity of one flux quantum (shown in Fig. 5) indicates that the sharp dips in the data truly correspond to a suppression of the signal periodically as a function of flux.

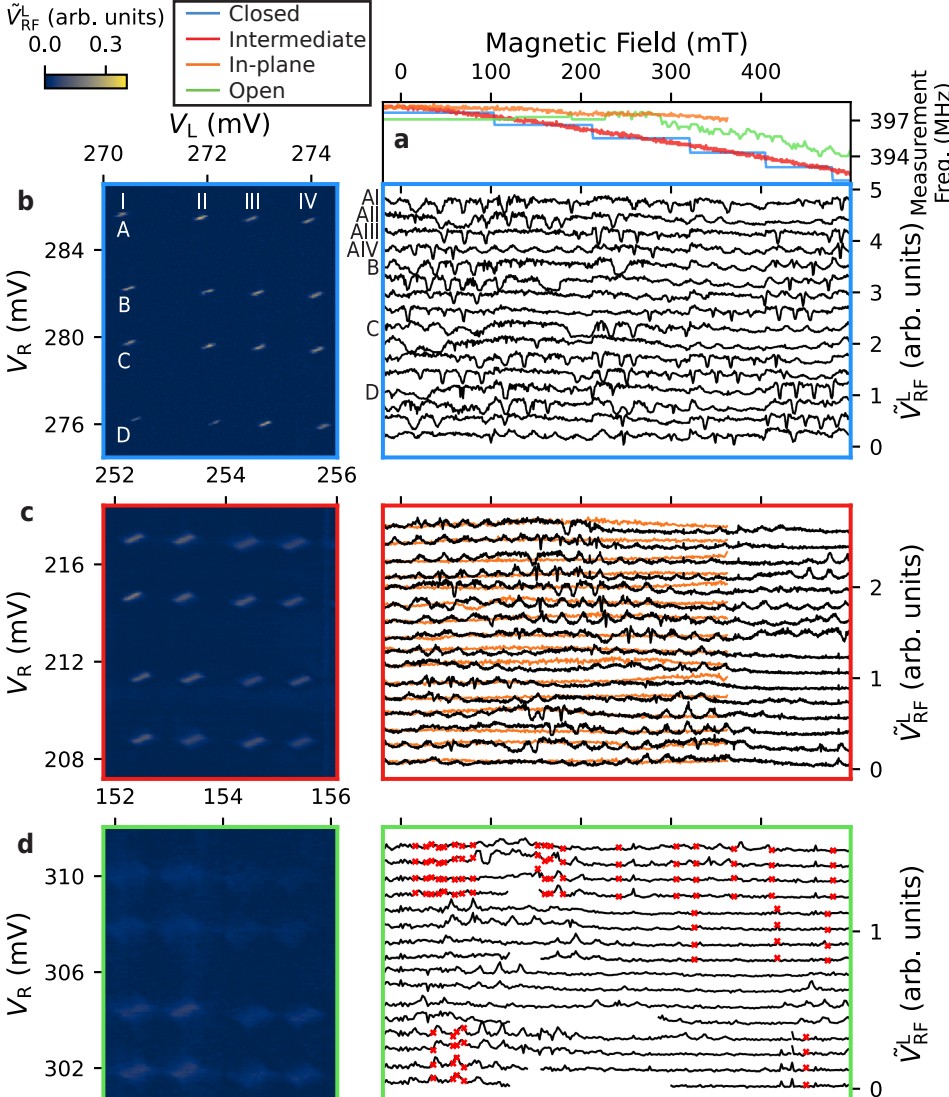

Figure 11: Field-dependence of inter-dot charge transitions in different regimes of tunnel coupling. **(a)** Measurement frequency for resonator L used at each out-of-plane field value $B_\perp$ for the three different regimes of tunneling strength investigated as well as for an in-plane field $B_\parallel$ sweep for the same transitions of the intermediate regime. **(b-d)** Field-dependence data for the closed **(b)**, intermediate **(c)**, and open **(d)** tunnel coupling regimes. These correspond to voltages $(V_{BT}, V_{BB}) = (-2.1, -1.65)$V, $(-1.9, -1.49)$V, $(-1.82, -1.34)$V, for the closed, intermediate, and open regimes respectively. $V_{BS}$ and $V_{BD}$ were tuned to a very weak tunneling regime of $V_{BS} = -2.05$ V and $V_{BD} = -2.75$ V, except in the closed regime where $V_{BS} = -2.5$ V. *Left:* CSDs measured at zero magnetic field, plotting the reflected signal magnitude $\tilde{V}_{RF}^L$ from resonator L centered about the Coulomb blockade value. *Right:* Field dependence of the peak deviation from Coulomb blockade for the 16 inter-dot transitions shown in the CSDs, offset by 0.3 **(b)**, 0.17 **(c)**, and 0.09 arb. units **(d)** for clarity. Peak heights in **(c)** for the $B_\parallel$ sweep are plotted in orange. In **(d)**, a stray resonance appeared which occluded inter-dot transitions for some transitions in a wide window. This resonance interfered with extraction of the peak signal height, and so appears as a gap in the plot. Red markers denote points at which charge jumps appeared in the search window used to extract the peak signal height.

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
