# Peer review of "Flux-Tunable Hybridization in a Double Quantum Dot Interferometer"

_SciPost Physics, doi:SciPost Phys. 17, 074 (2024)_

## Round 1 · Referee Report · Anonymous (Referee 1) · 2024-7-25

Strengths

1- Clear presentation of results and underlying physics 2- High quality data with clear interpretation 3- Demonstrating a proof-principle-measurement relevant for readout of topological qubits

Weaknesses

1- The system is relatively simple and behaves as expected.

Report

The authors present a quantum interference measurement of a circular double quantum dot probed via dispersive gate sensing. While there is a long history of measuring interferometers in mesoscopic devices, to my knowledge, this is the first demonstration of the interference in an isolated double quantum dot where readout needs to be performed by RF rather than transport measurements.

The overall setup is relatively simple (a ring-shaped double quantum dot with threaded flux). On one hand this allows for an excellent demonstration of an "irreducibly simple system where quantum interference is expected to occur", on the other hand the results are large exactly showing what one would expect out of such a simple system. What makes this paper more than only a pedagogical demonstration is the fact that the authors show that this simple system can actually be realized in a sizable mesoscopic device (micrometer long quantum dots) and that readout can be realized with dispersive gate sensing. Moreover, they explain some interesting subtleties in the level structure of the quantum dots. These findings are particularly relevant for the interferometric readout of topological qubits where similar quantum dot loops play a key role in many readout proposals.

Given this synergetic link and the clear path forward for more measurements in this direction I recommend the publication of the manuscript.

Requested changes

The timeliness of this paper is emphasized by the recently uploaded preprint arXiv:2401.09549 which uses similar measurements to read out the parity in an InAs-Al hybrid device. While the original arXiv posting of this manuscript clearly precedes this more recent preprint, I would assume that at the time of this review process a reference (or note) to this related preprint would be helpful for a reader.

Beyond that I have a few minor comments: - Fig. 8: While it can be calculated given the bias, it would be useful to explicitly state the conductance values of the three regimes in 8b. How does open/intermediate/closed relate to e^2/h? - App E: It is mentioned that $\tilde{\alpha}_i$ is different from the Coulomb blockade $\alpha_i$ due to the inter-dot capacitance. How important is that effect? What is the typical relative change of $\alpha$? Also, is it clear that an apparent difference is indeed due to the renormalization described in App F or could it also be due to uncertainty in the value of $L$ used to find $C_{\rm p}$? - App E: It wasn't quite clear to me how $f_0$ is extracted in order to find $C_{\rm p}$. Do the authors perform a frequency sweep of the RF tone for each value of $V_L$, $V_R$ and $B_\perp$ in Fig 3 to determine $f_0$? - App G: What are typical values of $\delta V_{\rm RF}$? Given that, how does the LZT condition compare to the measured values of $t_{\rm eff}$?

Recommendation

Publish (easily meets expectations and criteria for this Journal; among top 50%)

---

## Round 1 · Referee Report · Anonymous (Referee 2) · 2024-8-5

Report

The manuscript"Flux-Tunable Hybridization in a Double Quantum Dot Interferometer" by Christian G. Prosko et al. investigates the phenomenon of quantum interference in a system comprising two elongated quantum dots (QDs) arranged in a loop and subjected to a magnetic flux. The authors use radio-frequency reflectometry to measure the inter-dot tunnel coupling, discovering that this coupling oscillates with a periodicity of one flux quantum. These oscillations are influenced by the specific electronic levels involved in tunneling, and while the tunnel coupling amplitude varies, it does not completely diminish.

The manuscript's findings are generally interesting. Specifically, the ability to control and measure the hybridization between fermionic levels using magnetic flux is essential for implementing qubits with enhanced readout sensitivity. The results suggest that while it is possible to achieve a tunable coupling, the variability and partial suppression of coupling at oscillation minima present challenges. These insights are crucial for improving the design of qubit systems that exploit flux-tunable couplings, thereby contributing to the broader field of topological quantum computation.

In this regard, I can recommend the manuscript for SciPost. However, I still have the following comments:

(1) I wonder if the resonator survives at high magnetic fields?
(2) Throughout the manuscript, there are places where 'maximum' and 'minimum' are mentioned. These 'maximum' and 'minimum' are often referred to in different contexts compared to the previous sentence. Although they are well defined and make the manuscript precise, it makes the text rather hard to follow. For example, right after equation (1), 'Frequency shift on QDL’s gate resonator with a maximal value in the ground state.' It's unclear what 'maximum' here means. I guess this refers to 'the frequency shift due to tunnel coupling alone without other influence'.
(3) This comment is more about writing: On page 2, the author mentioned 'Coulomb diamond measurements demonstrate a varying but finite level spacing above 70 µeV in both QDs, such that the DQD is well-described by two coupled fermionic levels'. This conflicts with the potential spin degeneracy mentioned later. The influence of degenerate/excited states should be mentioned.

Recommendation

Publish (meets expectations and criteria for this Journal)

---

## Editorial Decision

published